# CK2 Activity Mediates the Aggressive Molecular Signature of Glioblastoma Multiforme by Inducing Nerve/Glial Antigen (NG)2 Expression

**DOI:** 10.3390/cancers13071678

**Published:** 2021-04-02

**Authors:** Beate M. Schmitt, Anne S. Boewe, Claudia Götz, Stephan E. Philipp, Steffi Urbschat, Joachim Oertel, Michael D. Menger, Matthias W. Laschke, Emmanuel Ampofo

**Affiliations:** 1Institute for Clinical & Experimental Surgery, Saarland University, 66421 Homburg, Germany; beate.schmitt@uks.eu (B.M.S.); anne.boewe@uks.eu (A.S.B.); michael.menger@uks.eu (M.D.M.); matthias.laschke@uks.eu (M.W.L.); 2Medical Biochemistry and Molecular Biology, Saarland University, 66421 Homburg, Germany; claudia.goetz@uks.eu; 3Experimental and Clinical Pharmacology and Toxicology, Center for Molecular Signaling (PZMS), Saarland University, 66421 Homburg, Germany; stephan.philipp@uks.eu; 4Department of Neurosurgery, Faculty of Medicine, Saarland University, 66421 Homburg, Germany; steffi.urbschat@uks.eu (S.U.); joachim.oertel@uks.eu (J.O.)

**Keywords:** glioblastoma multiforme, GBM, nerve/glial antigen 2, NG2, CK2, CX-4945, proliferation, migration, CRISPR/Cas9

## Abstract

**Simple Summary:**

Glioblastoma multiforme (GBM) is the most common and lethal primary malignant cancer of the central nervous system with a median patient survival of ~15 months. It has been reported that particularly nerve/glial antigen (NG)2-positive GBM is associated with an aggressive clinical phenotype and poor prognosis. Based on our latest findings, that protein kinase CK2 is a crucial regulator of NG2 expression in pericytes, we investigated the effect of CK2 inhibition by CX-4945 as well as CK2 KO on NG2 expression in human GBM cells. We found that CK2 inhibition suppresses proliferation and migration of different NG2-positive GBM cells. In silico analyses revealed a positive correlation between the mRNA expression of the two proteins. Moreover, we verified the decreased expression of NG2 in patient-derived GBM cells after CX-4945 treatment. These novel insights into the molecular signaling of NG2-positive GBM demonstrate that CX-4945 may represent a promising drug for future GBM therapy.

**Abstract:**

Nerve/glial antigen (NG)2 expression crucially determines the aggressiveness of glioblastoma multiforme (GBM). Recent evidence suggests that protein kinase CK2 regulates NG2 expression. Therefore, we investigated in the present study whether CK2 inhibition suppresses proliferation and migration of NG2-positive GBM cells. For this purpose, CK2 activity was suppressed in the NG2-positive cell lines A1207 and U87 by the pharmacological inhibitor CX-4945 and CRISPR/Cas9-mediated knockout of CK2α. As shown by quantitative real-time PCR, luciferase-reporter assays, flow cytometry and western blot, this significantly reduced NG2 gene and protein expression when compared to vehicle-treated and wild type controls. In addition, CK2 inhibition markedly reduced NG2-dependent A1207 and U87 cell proliferation and migration. The Cancer Genome Atlas (TCGA)-based data further revealed not only a high expression of both NG2 and CK2 in GBM but also a positive correlation between the mRNA expression of the two proteins. Finally, we verified a decreased NG2 expression after CX-4945 treatment in patient-derived GBM cells. These findings indicate that the inhibition of CK2 represents a promising approach to suppress the aggressive molecular signature of NG2-positive GBM cells. Therefore, CX-4945 may be a suitable drug for the future treatment of NG2-positive GBM.

## 1. Introduction

CK2 is a ubiquitously expressed, constitutively active serine/threonine kinase consisting of two catalytic CK2α- or CK2α’- and two non-catalytic CK2β-subunits [1]. With more than 500 substrates, CK2 is involved in various biological processes and estimated to be responsible for up to 10% of the human phosphoproteome [2,3]. CK2 exerts oncogenic activity, because its overexpression promotes tumor development and progression via the activation of proliferation and inhibition of apoptosis [4,5,6]. Hence, a broad spectrum of CK2 inhibitors have been developed as potential anti-cancer drugs, including CIGB-300 [7], 4,5,6,7-tetrabromobenzotriazole (TBB) [8] and CX-4945 [9].

Nerve/glial antigen (NG)2 is a type-1 transmembrane proteoglycan with a core of 290 kDa that is expressed in pericytes and different progenitor cells under physiological conditions [10]. The extracellular domain of the proteoglycan is able to bind components of the extracellular matrix (ECM) [11] as well as to interact with integrin β1, platelet-derived growth factor (PDGF)α and fibroblast growth factor (FGF)2 [12]. These interactions trigger different signaling pathways, including extracellular signal-regulated kinases (ERK)1/2 and focal adhesion kinase (FAK) [13,14]. This, in turn, activates cell proliferation, cell motility, inflammation and angiogenesis [15,16,17,18]. Recently, we have identified CK2 as a novel regulator of NG2-dependent signaling pathways [19]. Our results revealed that inhibition of CK2 suppresses NG2 expression in pericytes, which reduces their angiogenic activity [19].

Besides its expression in normal tissue, NG2 is also expressed in distinct tumors, including glioblastoma multiforme (GBM). GBM is categorized as grade IV glioma and associated with a poor outcome, as reflected by a median survival of less than 15 months from the day of diagnosis [5,20,21]. Although considerable efforts have been made in GBM research, effective therapeutic approaches for the treatment of this cancer type are still missing [21]. This is due to the fact that GBM expresses various genes promoting proliferation, invasion as well as drug resistance of tumor cells [5,21,22]. Several studies reported that particularly NG2-positive GBM is associated with an aggressive clinical phenotype and poor prognosis [11,23,24,25], which is why the proteoglycan is described as a potential therapeutic target.

Based on these findings, we hypothesize that CK2 inhibition suppresses proliferation and migration of NG2-positive GBM cells. To test this, we investigated the effect of pharmacological CK2 inhibition with CX-4945 as well as CK2α knockout (KO) on NG2 expression, cell proliferation and migration in human NG2-positive GBM cell lines. The Cancer Genome Atlas (TCGA)-based data from human gliomas were used to study the mRNA expression of CK2 and NG2 as well as the correlation between the two proteins. Finally, we assessed the expression of NG2 in patient-derived GBM cells, which were treated with CX-4945.

## 2. Results

### 2.1. CK2 Inhibition Reduces NG2 Expression in Human GBM Cell Lines

First, we investigated the effect of CK2 inhibition on NG2 protein expression in the NG2-positive human GBM cell lines A1207 and U87. For this purpose, the cells were treated with the CK2 inhibitor CX-4945 (10 µM) for 72 h and the expression of different proteins was assessed by western blot and flow cytometry. We found that all CK2 subunits are expressed in A1207 and U87 cells (Figure 1A and Appendix A). Of note, the expression of CK2α was more pronounced when compared to that of CK2α’ (Figure 1A and Appendix A). In addition, we detected a significantly reduced phosphorylation of the CK2 specific phosphorylation site serine 129 of Akt (pAkt^S129^) in CX-4945-treated cells. This confirms the efficiency of the CK2 inhibitor (Figure 1A–C and Appendix A). Moreover, CX-4945 significantly reduced the NG2 expression in the two cell lines when compared to controls (Figure 1A,D–G and Appendix A). To exclude that the herein observed effects of CX-4945 are independent of CK2 inhibition, we additionally generated CK2α KO in A1207 and U87 cells by means of the CRISPR/Cas9 system. CK2α was completely absent in KO cells. The protein expression of CK2β was markedly reduced (Figure 1H and Appendix A). This observation has already been reported for other cell lines and can be explained by the instability of free CK2β subunits [26]. As expected, the loss of CK2α resulted in a markedly reduced phosphorylation of Akt^S129^ and expression of NG2 (Figure 1H–N and Appendix A). These results were verified by western blot analyses of additional CK2α KO clones (Appendix A). Moreover, we noticed a decreased phosphorylation of FAK in CK2α KO cells when compared to wild type cells (Figure 1O,P and Appendix A).

### 2.2. CK2 Inhibition Reduces NG2 Gene Expression in Human GBM Cell Lines

Recently, we have identified a 114-bp fragment (NG2^p1.2.4.1^) close to the NG2 start codon as a CK2-dependent active promoter region in human pericytes [19]. Therefore, we investigated whether the activity of NG2^p1.2.4.1^ is also affected by CK2 inhibition in GBM cells. To test this, A1207 CK2α KO and wild type cells were transfected with pGL4-NG2^p1.2.4.1^ reporter construct and the luciferase activity was measured (Figure 2A). In addition, A1207 cells transfected with pGL4-NG2^p1.2.4.1^ reporter construct were treated with CX-4945 or vehicle and subsequently analyzed by luciferase assay (Figure 2B). In line with our previous results, we detected a reduced transcriptional activity of NG2^p1.2.4.1^ after CK2 inhibition (Figure 2A,B). Additional quantitative real-time PCR (qRT-PCR) analyses showed a reduced NG2 mRNA expression in CX-4945-treated A1207 and U87 cells when compared to controls (Figure 2C,D). We further performed cycloheximide experiments to study the stability of the NG2 protein. Flow cytometric analyses clearly demonstrated that CX-4945 does not affect the stability of the protein (Figure 2E,F).

### 2.3. CK2 Inhibition Suppresses the Proliferation of Human GBM Cell Lines

In the next set of experiments, we assessed the proliferation of NG2-positive GBM cell lines after CX-4945 treatment. Water-soluble tetrazolium (WST)-1 assay revealed a significantly reduced mitochondrial activity in A1207 and U87 cells after CX-4945 treatment when compared to vehicle-treated controls (Figure 3A,B). Additional flow cytometric analyses of Bromodeoxyuridine (BrdU) incorporation, growth curves and scratch assays of CX-4945-and vehicle-treated cells confirmed the anti-proliferative effect of CK2 inhibition (Figure 3C–J). Of note, lactate dehydrogenase (LDH) assays showed that this effect is not caused by cytotoxicity of 10 µM CX-4945 (Figure 3K,L).

### 2.4. CK2 Inhibition Reduces the Migratory Capacity of NG2-Positive GBM Cell Lines

NG2 promotes cell migration via its binding to the ECM [11,23,24]. To study the impact of CK2 on NG2-dependent cell migration, we performed a panel of transwell assays. As expected, silencing of NG2 in A1207 cells markedly reduced the number of migrated cells (Figure 4A–C). Furthermore, CK2 inhibition with CX-4945 also reduced the migratory capacity of A1207 cells (Figure 4D,E). To verify that CK2 mediates cell migration via NG2, we overexpressed NG2 in A1207 cells (Figure 4F and Appendix A), which were subsequently treated with CX-4945. We found that overexpression of NG2 partially rescued the reduced migratory capacity of CX-4945-treated cells (Figure 4G,H).

### 2.5. CK2 and NG2 mRNA Expression Positively Correlate in Human GBM

Next, we assessed the gene expression of CK2 subunits and NG2 in different human gliomas (grade I–IV) based on TCGA data. These analyses showed that the mRNA expression of NG2 is significantly elevated in GBM compared to lower grade gliomas (Figure 5A). Of interest, the mRNA expression of CK2α, CK2α’ and CK2β was also significantly higher in GBM (Figure 5B–D). We noticed positive correlations between all CK2 subunits and NG2 in GBM (Figure 5E–G). Particularly, we detected a strong positive correlation between CK2α and NG2, as indicated by a Spearman’s correlation coefficient of r = 0.54 (*p* < 0.0001). These findings support our in vitro results showing that CK2 is involved in NG2 expression.

### 2.6. CK2 Inhibition Reduces NG2 Expression in Patient-Derived GBM Cells

Finally, we investigated the effect of CX-4945 on NG2 expression in patient-derived GBM cells. The herein analyzed cells (T8399, T8478, T8475 and T8470) originated from primary GBM grade IV, as indicated by a lack of the *IDH1* mutation R132H [27]. Of note, NG2 and the two catalytic CK2α and α’ subunits were expressed in all patient-derived cells. Although the expression of these proteins markedly differed between the cells of individual patients, CX-4945 treatment reduced the CK2-dependent phosphorylation of Akt^S129^ as well as NG2 expression in all patient-derived cells (Figure 6A–D and Appendix A).

## 3. Discussion

In the present study, we hypothesized that CK2 inhibition suppresses proliferation and migration of NG2-positive GBM cells. Of interest, we found that the pharmacological inhibition as well as CK2α KO in NG2-postive GBM cell lines significantly reduces NG2 expression. Gene regulatory analyses further demonstrated that this is due to a diminished gene expression of NG2. Moreover, we detected a markedly decreased NG2-dependent cell proliferation and migration after CK2 inhibition. TCGA-based data revealed that both NG2 and CK2 are highly expressed in GBM. In addition, we observed a positive correlation between the mRNA expression of CK2α and NG2. Finally, we verified the decrease of NG2 expression after CX-4945 treatment in patient-derived GBM cells. This indicates that the oncogenic activity of CK2 mediates the aggressive molecular signature of GBM by inducing NG2 expression.

The overexpression of CK2 in various tumor types suppresses apoptosis while promoting cell proliferation and migration by dysregulating signaling pathways, such as nuclear factor kappa (NFk)B and phosphoinositide 3-kinase (PI3K)/Akt. Given these central functions of CK2 in tumorigenesis, it is not surprising that a broad spectrum of CK2 inhibitors has been developed, culminating in the synthesis of CX-4945. This compound is the most specific CK2 inhibitor to date and has a high bioavailability [28]. Of note, CX-4945 has the ability to cross the blood-brain-barrier and, thus, may be suitable in the treatment of brain tumors, including GBM [29,30].

GBM is the most common and lethal primary malignant cancer of the central nervous system with a median patient survival of ~15 months from the day of diagnosis [1]. Investigating the molecular mechanisms of CK2 in GBM development and progression, the group of Benveniste demonstrated that this kinase is required for the activation of pro-survival pathways, such as Januskinase/Signal Transducers and Activators of Transcription (JAK/STAT) and PI3K/AKT [31,32]. Moreover, it is well known that GBM often exhibits a striking cellular heterogeneity [20,33,34]. For instance, it has been shown that NG2-positive cells are associated with an aggressive clinical phenotype and poor prognosis [11,23,24,25]. We herein found that the inhibition of CK2 significantly reduces NG2 gene expression in the GBM cell lines A1207 and U87. For this purpose, CX-4945 was used to suppress CK2 activity. To identify suitable, non-toxic concentrations of this inhibitor, GBM cells were treated with 2.5 µM, 5 µM and 10 µM CX-4945 over 24 h, 48 h and 72 h, and both cell proliferation and cytotoxicity were determined. We found that none of the tested concentrations is cytotoxic. However, only 10 µM CX-4945 for 72 h lowered cell proliferation more than 50%. More importantly, 10 µM CX-4945 was the most efficient concentration for the reduction of NG2 expression. Based on these findings, we decided to use 10 µM CX-4945 to study the effects on NG2 expression and NG2-dependent cellular functions. To exclude that the herein observed reduced NG2 expression is due to unspecific effects of CX-4945 and, thus, CK2-independent, we additionally generated CK2α KO cell lines showing that the loss of CK2 activity significantly reduces NG2 expression. The analysis of the gene regulatory mechanism revealed that this is caused by a decreased transcriptional activity of a 114 bp fragment close to the NG2 start codon. This region harbors a binding site for the transcription factor SP1, which can activate or repress gene expression [35]. It has been reported that posttranslational modifications of SP1, including phosphorylation, affect GBM cell proliferation and invasion [36,37]. Of note, CK2 phosphorylates SP1 and inhibition of this kinase increases its DNA binding capacity [38,39]. Hence, it is tempting to speculate that CK2 regulates NG2 gene expression via SP1-dependent phosphorylation in GBM cells.

Various posttranslational modifications of the NG2 core protein, resulting in higher molecular forms, have been identified. For instance, the group of Stallcup showed that NG2 is phosphorylated at threonine 2256 by protein kinase C and threonine 2314 by ERK1/2, which promotes cell proliferation and cell motility [40]. Moreover, high molecular forms of NG2 caused by glycosylation are observed. However, the nature of this glycosylation and its biological significance is still unknown [41,42]. In this study, we detected higher molecular forms of NG2 by western blot, which are partially reduced after CK2 inhibition. Hence, we cannot exclude that CK2 additionally reduces the NG2 protein level by phosphorylation of the proteoglycan or by inhibition of glycosyltransferases.

In GBM, NG2 is involved in cell proliferation, migration and invasion via a wide range of molecular interactions [24]. The proteoglycan promotes cell proliferation and motility by binding to integrin β1, PDGFα and FGF2 [12]. In addition, the invasive and migratory activity of NG2-positive GBM cells is stimulated by the interaction with ECM proteins [11]. In line with these findings, we could show that CK2 inhibition reduces the migration of NG2-positive GBM cell lines. It is well known that CK2 regulates cell migration via various signaling pathways [31,32]. To verify that the herein observed anti-migratory effect is mediated by NG2, we performed additional rescue experiments. Our results clearly demonstrate the importance of NG2 in CK2-dependent migration of GBM cells, as shown by an improved migratory ability of NG2-overexpressing cells after CK2 inhibition.

FAK mediates integrin- and growth factor-induced signaling transduction resulting in the activation of proliferative pathways. Kim et al. [43] reported that CX-4945 is capable of reducing TGF-β1-induced FAK phosphorylation. Furthermore, silencing of NG2 decreases the activity of FAK, resulting in a diminished cell proliferation [44]. In our study, we observed a diminished phosphorylation of FAK in CK2α KO cells. Therefore, it is tempting to speculate that the reduced proliferation of NG2-positive, CX-4945-treated GBM cells may be caused by a disturbed NG2/FAK signaling.

Recently, Al-Mayhani et al. [23] reported that NG2-positive cells from GBM patient tumor samples proliferate faster than NG2-negative ones. On the other hand, it has been shown that NG2 promotes the vascularization of GBM [45]. This indicates that NG2 is not only involved in tumor cell proliferation but also in angiogenesis. We previously found that CK2 inhibition reduces the NG2-dependent angiogenic activity of human pericytes [19]. Therefore, we suggest that CX-4945 is a promising compound for the treatment of NG2-positive GBM, because it targets both the vascular and tumor compartment.

We further analyzed the mRNA expression pattern of NG2, CK2α, CK2α’ and CK2β in different gliomas using TCGA-based data. As previously reported [24,46], we found the highest expression of NG2 in GBM when compared to other gliomas. The expression of the CK2 subunits was also remarkably higher in GBM. The correlation between NG2 and CK2 mRNA revealed that all CK2 subunits were positively correlated with NG2. Of note, the strongest correlation was observed between CK2α and NG2. This may be explained by the fact that CK2α and CK2α’ exert tissue-specific functions, because of their partially different substrates [26]. We herein detected an increased expression of CK2α when compared to CK2α’ in NG2-positive GBM cells. This indicates that CK2α may have a superior function in the regulation of NG2 expression when compared to CK2α’.

Overexpression of NG2 has not been reported to be a result of genetic aberrations, such as gene amplifications or chromosome translocation, in GBM [47]. Moreover, there are no mutations known in the NG2 gene leading to gain or loss of function. In contrast, genome-wide copy number variation analyses in GBM demonstrated that chromosome 20 harbors frequent gains in gene dosage, which may be driven by several oncogenic targets [48]. Of note, the CK2α gene is located on chromosome 20 and it has been reported that CK2 expression is required for the activation of survival pathways, including the JAK/STAT, NFκB and PI3K/AKT pathways in GBM [32]. Hence, it can be assumed that the gains in CK2α gene dosage may be a crucial oncogenic driver during gliomagenesis.

Finally, we assessed the effect of CK2 inhibition on patient-derived NG2-positive GBM cells. This is of major importance, as recent evidence suggests that classically established cell lines from different tumors, including GBM, do not fully reflect the genotypes and phenotypes of primary tumors [49]. Notably, we also detected a decreased NG2 expression after CX-4945 treatment in patient-derived cells, demonstrating that our cell line-based results on CK2/NG2-interaction in GBM are robust and reproducible in clinical samples. CX-4945 is currently tested in phase I and II clinical trials for the treatment of different cancer types, including cholangiocarcinoma (NCT02128282), multiple myeloma (NCT01199718) and medulloblastoma (NCT03904862). Therefore, we suggest that treatment of NG2-positive GBM with this compound represents a promising therapeutic approach, which should be clinically evaluated in the near future.

## 4. Materials and Methods

### 4.1. Chemical and Biological Reagents

Roswell Park Memorial Institute (RPMI) 1640 medium, Dulbecco’s Modified Eagle’s Medium (DMEM), Lipofectamine3000 reagent, Opti-MEM Reduced Serum Medium (Gibco), fetal calf serum (FCS), penicillin-streptomycin and small interfering RNA (siRNA) duplexes directed against NG2 (ID: 146.147) were from Thermo Fisher Scientific (Karlsruhe, Germany). Cycloheximide was from Sigma-Aldrich (Taufkirchen, Germany). Bovine serum albumin (BSA) was from Santa Cruz Biotechnology (Heidelberg, Germany). CX-4945 was from ActivateScientific (Prien, Germany). BrdU was from Roche (Mannheim, Germany). Cell lysis reagent QIAzol and HiPerFect transfection reagent were from Qiagen (Hilden, Germany). The qScriber cDNA Synthesis Kit and ORA SEE qPCR Green ROX L Mix were from HighQu (Kraichtal, Germany). The NG2-plasmid (pEF6-CSPG4-myc-his) was from addgene (Watertown, MA, USA). Luciferase Assay System was from Promega (Walldorf, Germany).

### 4.2. Antibodies

Anti-NG2 antibody (sc-166251) and anti-CK2β antibody (E9) were from Santa Cruz Biotechnology. Anti-β-actin antibody (66009) and anti-α-tubulin antibody (66031) were from Proteintech Germany GMBH (St. Leon-Rot, Germany). The anti-Akt1/2/3 antibody (11E7), anti-FAK antibody (3285) and anti-pFAK antibody (3283) were from Cell Signaling (Frankfurt am Main, Germany). Anti-CK2α antibody and anti-CK2α’ antibody were generated as described previously [50]. Anti-pAKT antibody (EPR6150) was from Abcam (Cambridge, UK). Peroxidase-labeled anti-mouse antibody (NIF 825) and peroxidase-labeled anti-rabbit antibody (NIF 824) were from GE healthcare (Freiburg, Germany). Anti-chondroitin sulfate proteoglycan 4 (NG2) (562415) was from BD Biosciences (Heidelberg, Germany) and the BrdU antibody (BU20A) was from eBioscience Fisher Scientific (Schwerte, Germany).

### 4.3. Cell Culture

The human GBM cell lines A1207 (SymbioTec GmbH, Saarbrücken, Germany) and U87 (ATCC, Manassas, VA, USA) were cultivated in RPMI or DMEM supplemented with 10% FCS and penicillin-streptomycin at 37 °C under a humidified 95%/5% (vol/vol) mixture of air and CO_2_. The cells were passaged at a split ratio of 1:3 after reaching confluence.

Primary GBM cells (T8399, T8478, T8475 and T8470) were obtained from the tumor tissue of patients undergoing surgery at the Department of Neurosurgery (Saarland University), as previously described in detail [51]. Briefly, tumor samples were mechanically processed and the resulting cell suspension was cultured in DMEM supplemented with 10% FCS, 1% non-essential amino acids and 1% penicillin-streptomycin at 37 °C and 5% CO_2_. The medium was changed twice a week and the cells were used at passage 0 to 2. The study was approved by the local German ethical board (Ethikkommission der Ärztekammer des Saarlandes, Saarbrücken, Germany, General Medical Council of the State Saarland, NO 93/16).

### 4.4. WST-1 Assay

A WST-1 assay (Roche) was used to determine the mitochondrial activity of A1207 and U87 cells as a parameter of cell viability, as described previously in detail [52].

### 4.5. LDH Assay

A LDH assay (Cytotoxicity Detection KitPLUS, Roche) was used to evaluate the cytotoxic effects of CX-4945 on A1207 and U87 cells, as previously described in detail [52].

### 4.6. Generation of CK2α KO Cells by CRISPR-Cas9

All-in-one plasmid expressing Cas9-Dasher green fluorescent protein (GFP) and the two single guide RNAs (sgRNA) (pD 1401-AD: CMV-Cas9N-2A-GFP, Cas9-ElecD) to target CK2α were from ATUM (Newark, CA, USA). The sgRNA guide sequences targeting CK2α were: 5′-CCTGGATTATTGTCACAGCA-3′ and 5′-GGTGGGATGAACGGGTCAGAA-3′. The CK2α KO was performed as previously described in detail [53]. Briefly, cells were transfected with the all-in-one plasmid by means of Lipofectamine3000 according to the manufacturer’s instructions. Forty-eight hours after transfection, single GFP-positive cells were separated by fluorescence-activated cell sorting (MoFlo XDP Cell Sorter (Beckman Coulter GmbH, Krefeld, Germany) and expanded to obtain individual clones. CK2α KO was verified by western blot analysis.

### 4.7. Reporter Gene Assay

The transcriptional activity of the NG2 promoter fragment NG2^p1.2.4.1^ was assessed by reporter gene assays according to the manufacturer’s instructions (Promega). Briefly, A1207 cells were transfected with pGL4 or pGL4-NG2^p1.2.4.1^ reporter vectors by Lipofectamine3000 for 24 h. The cells were lysed and the luciferase activity was detected by a luminescence plate reader.

### 4.8. NG2 Silencing and NG2 Overexpression

For NG2 silencing, A1207 cells were transfected using HiPerFect transfection reagent for 72 h with 20 nM control siRNA or NG2 siRNA: sense 5′-GCUAUUUAACAUGGUGCUGtt-3′ and antisense 5′-CAGCACCAUGUUAAAUAGCtt-3′. For NG2 overexpression, the cells were transfected with a ctrl-plasmid (mock) or pEF6-CSPG4-myc-his using Lipofectamine3000 reagent for 48 h. Subsequently, the cells were harvested and used for flow cytometry or western blot analyses.

### 4.9. Scratch Assay

The migratory capacity of A1207 and U87 cells was assessed by means of scratch assays. The cells were seeded in a 24-well plate and cultivated until confluence. Subsequently, the cell monolayer was scratched with a pipette tip and then washed twice with phosphate-buffered saline (PBS) to remove non-adherent cells. Phase-contrast light microscopic images were taken immediately (0 h) and 24 h after scratching. The gap area was determined by means of ImageJ software (U.S. National Institutes of Health (NIH), Bethesda, MD, USA).

### 4.10. Growth Curves

A1207 and U87 cells were seeded in a 24-well plate and cultivated for 16 h. Thereafter, the cells were treated with vehicle or CX-4945 (10 µM) for 24 h, 48 h and 72 h. Then, the cells were detached, centrifuged and suspended with fresh culture medium. Ten µL of the suspended cells were stained with trypan blue solution (0.4%) and counted by a LUNA Automated Cell Counter according to the manufacturer’s protocol.

### 4.11. Transwell Migration Assay

The migratory activity of A1207 cells was analyzed using 24-well chemotaxis chambers and polyvinylpyrrolidone-coated polycarbonate filters with a pore size of 8 µm (BD Biosciences). The filters were incubated overnight (37 °C, 5% CO_2_) in RPMI without any supplements, medium was removed and 750 µL culture medium supplemented with 5% FCS was added to each of the lower wells. The upper wells were filled with 200 µL RPMI (0.1% FCS) containing 2.0 × 10^5^ treated cells. Non-migrated cells were removed from the upper surface of the filters by cotton swabs after 5 h of cultivation. Migrated cells, which adhere to the lower surface, were fixed with methanol and stained with Dade Diff-Quick (Dade Diagnostika GmbH, München, Germany). Migrated cells were counted in 20 microscopic high-power fields (HPF) at 200× magnification (BZ-8000; Keyence, Osaka, Japan).

### 4.12. Western Blot Analysis

The separation of whole cell extracts was performed through a 7.5% or 12.5% SDS polyacrylamide gels, which were transferred onto a polyvinylidene difluoride (PVDF) membranes. The membranes were blocked with 5% BSA in Tris-buffered saline TBS (0.1% Tween20) for 1 h and subsequently incubated with the primary antibodies (anti-pAkt, anti-Akt, anti-pFAK, FAK, anti-CK2α, anti-CK2α’, anti-CK2β, anti-NG2, anti-β-actin and anti-α-tubulin; 1:500) in TBS (0.1% Tween20, 1% BSA) overnight (4 °C). The membrane was incubated with a peroxidase-coupled secondary antibody (anti-rabbit 1:1500 or anti-mouse 1:2000) for 1 h and washed with TBS (0.1% Tween20, 1% BSA). The expression of the proteins was visualized by luminol-enhanced chemiluminescence (ECL; GE Healthcare).

### 4.13. Flow Cytometry

Cells were washed with PBS (4 °C) and harvested by scratching. Subsequently, the cells were incubated with a phycoerythrin (PE)-labeled primary anti-NG2 antibody for 1 h at room temperature. Afterwards, the cells were washed in PBS and the MFI of 3000 cells was analyzed by a FACSLyrics flow cytometer (BD).

Cell proliferation was assessed by a BrdU-assay according to the manufacturer’s protocol (Thermo Scientific). Briefly, A1207 and U87 cells were incubated with BrdU. After 10 h, the cells were washed, fixed and permeabilized. Incorporated BrdU was detected by using a BrdU antibody. The MFI of 3000 cells was analyzed in the FL-1 and FL-2 channel by a FACSLyrics flow cytometer.

### 4.14. Gene Expression Analysis

QIAzol lysis reagent was used to isolate total RNA and cDNA was transcribed by means of a qScriber cDNA Synthesis Kit. The amount of mRNA was determined by qRT-PCR using ORA SEE qPCR Green ROX L Mix according to the manufacturer’s instructions. Primers (NG2 forward 5′-GGCTGTCAAAACCAGGGTAA-3′ and reverse 5′-AGAGGGCAAGGAGAAGGAAG-3′; GAPDH forward 5′-CCACCCATGGCAAATTCC-3′ and reverse 5′-ACTCCACGACGTACTCAG-3′) were used at a concentration of 500 nM. Data collection and analyses were performed by a MiniOpticon Real-Time PCR Detection System and the 2^–ΔΔCt^ method.

### 4.15. TCGA Data-Based Analyses

We used two TCGA datasets (Firehose Legacy (Brain Lower Grade Glioma) and Cell 2013 (Glioblastoma)) from cBioPortal (http://cbioportal.org, accessed on 10 November 2020). This platform is an open-access resource for interactive analyses of multidimensional cancer genomics datasets [54,55], to analyze the mRNA expression of NG2, CK2α, CK2α’ and CK2β in different human gliomas (human astrocytoma (grade I-II), oligodendroglioma (grade II), anaplastic oligoastrocytoma and astrocytoma (grade III) as well as in GBM (grade IV)). Moreover, we used RNA Seq V2 RSEM-based data from the TCGA dataset (Cell 2013 (Glioblastoma)) to analyze the correlations between NG2 and CK2 subunits.

### 4.16. Statistical Analysis

All data were tested for normal distribution and equal variance. Differences between two groups were assessed by the unpaired Student’s *t*-test. To detect differences between multiple groups, one-way ANOVA was applied. This was followed by the Tukey post-hoc test, including the correction of the α-error according to Bonferroni probabilities. The association between NG2 and CK2 subunits was analyzed using the Spearman’s correlation method. Statistics were performed by GraphPad Prism (version 8). All values are given as mean ± SD. Statistical significance was accepted for *p* < 0.05.

## 5. Conclusions

The expression of NG2 in GBM is associated with an aggressive clinical phenotype and poor prognosis. In the present study, we identified the protein kinase CK2 as a novel regulator of NG2 expression in GBM. We could demonstrate that inhibition of this kinase by CX-4945 as well as CK2 knockout significantly decreases the expression of NG2, resulting in a reduced proliferation and migration of GBM cells. In silico, analyses showed a positive correlation between CK2 and NG2 in TCGA-based data. More importantly, we detected a reduced NG2 expression after CX-4945 treatment in patient-derived GBM cells. In conclusion, inhibition of CK2 activity may represent a promising approach for NG2-positive GBM therapy.

## Figures and Tables

**Figure 1 cancers-13-01678-f001:**
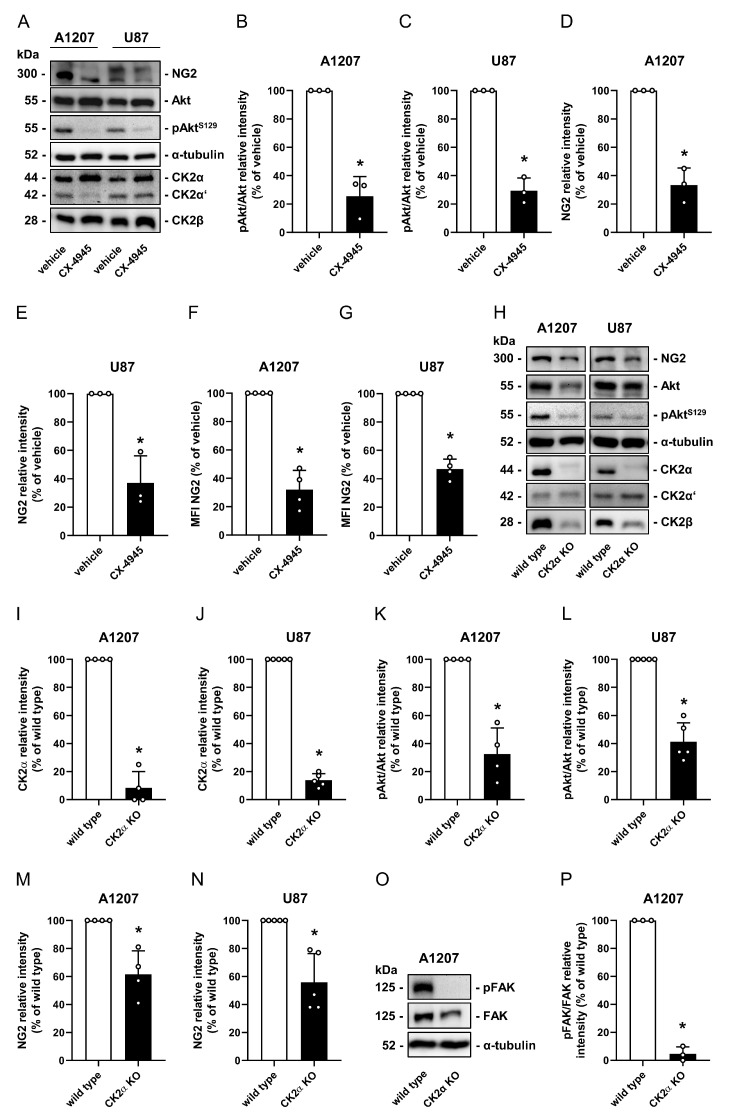
CK2 inhibition reduces NG2 expression in human GBM cell lines. (**A**) A1207 and U87 cells were treated with vehicle (DMSO) or CX-4945 (10 µM) for 72 h. The cells were lysed and the expression of NG2, Akt, pAkt^S129^, CK2α, CK2α’, CK2β and α-tubulin (as loading control) was analyzed by western blot. (**B**–**E**) A1207 and U87 cells were treated as described in (**A**) and the expression of pAkt/Akt (**B**,**C**) and NG2 (**D**,**E**) was quantitatively analyzed. Vehicle-treated cells were set 100%. Mean ± SD. * *p* < 0.05 vs. vehicle (*n* = 3). (**F**,**G**) A1207 and U87 cells were treated as described in (**A**), scratched and the mean fluorescence intensity (MFI) of NG2-positive cells was assessed by flow cytometry. The MFI of vehicle-treated cells was set 100%. Mean ± SD. * *p* < 0.05 vs. vehicle (*n* = 4). (**H**) A1207 and U87 wild type and CK2α KO cells were lysed and the expression of NG2, Akt, pAkt^S129^, CK2α, CK2α’, CK2β and α-tubulin (as loading control) was analyzed by western blot. (**I**–**N**) A1207 and U87 cells were treated as described in (**H**) and the expression of CK2α (**I**,**J**), pAkt/Akt (**K**,**L**) and NG2 (**M**,**N**) was quantitatively assessed. Wild type cells were set 100%. Mean ± SD. * *p* < 0.05 vs. wild type (A1207: *n* = 4; U87: *n* = 5). (**O**) A1207 wild type and CK2α KO cells were lysed and the expression of FAK, pFAK and α-tubulin (as loading control) was analyzed by western blot. (**P**) A1207 wild type and CK2α KO cells were lysed and the expression of pFAK/FAK was quantitatively assessed. Wild type cells were set 100%. Mean ± SD. * *p* < 0.05 vs. wild type (*n* = 3).

**Figure 2 cancers-13-01678-f002:**
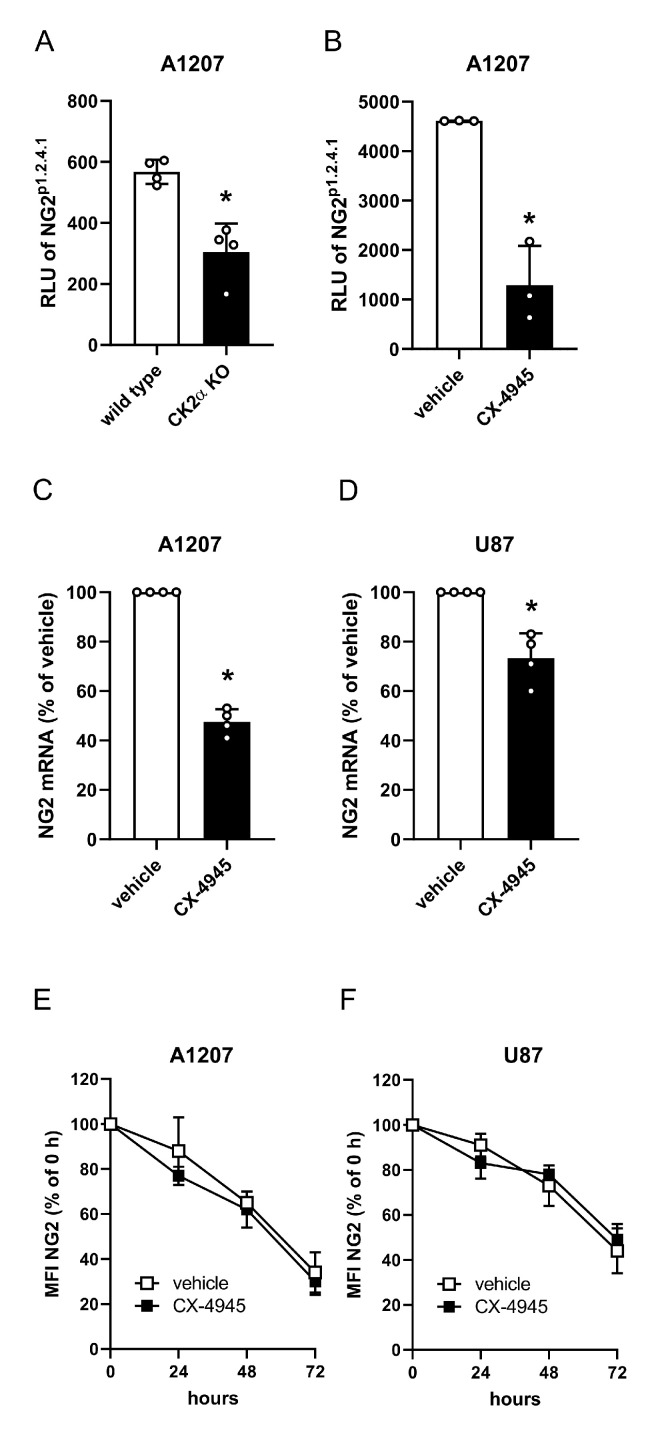
CK2 inhibition reduces NG2 gene expression in human GBM cell lines. (**A**) A1207 wild type and CK2α KO cells were transfected with pGL4-NG2^p1.2.4.1^, cultivated for 24 h, lysed and the transcriptional activity was detected by a luciferase assay. Relative luciferase units (RLU) of wild type cells were used as control. Mean ± SD. * *p* < 0.05 vs. wild type (*n* = 4). (**B**) A1207 cells were transfected with pGL4-NG2^p1.2.4.1^ for 24 h, treated with vehicle (DMSO) or CX-4945 (10 µM) and analyzed by a luciferase assay. RLU of vehicle-treated cells were used as control. Mean ± SD. * *p* < 0.05 vs. vehicle (*n* = 3). (**C**,**D**) A1207 and U87 cells were treated with vehicle (DMSO) or CX-4945 (10 µM) for 72 h, harvested and total RNA was isolated. The relative gene expression of NG2 was examined by qRT-PCR normalized to GAPDH. Vehicle-treated cells were set 100%. Mean ± SD. * *p* < 0.05 vs. vehicle (*n* = 4). (**E**,**F**) A1207 and U87 cells were treated with vehicle (DMSO) or CX-4945 (10 µM) in the presence of cycloheximide. The cells were scratched after 0 h, 24 h, 48 h and 72 h and the MFI of NG2-positive cells was assessed by flow cytometry. MFI of cells at 0 h was set 100%. Mean ± SD (*n* = 4).

**Figure 3 cancers-13-01678-f003:**
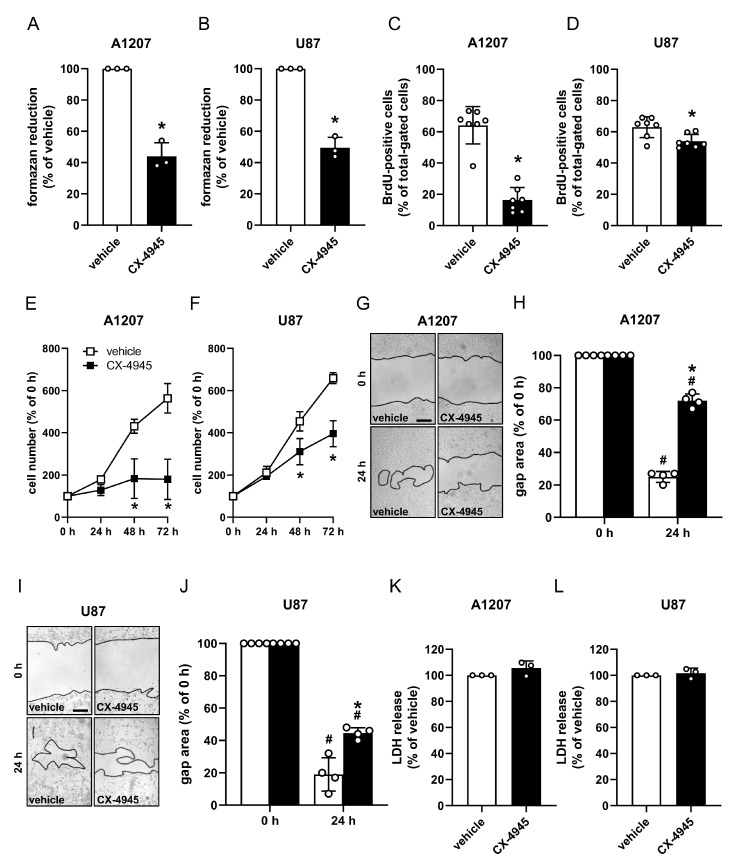
CK2 inhibition suppresses the proliferation of human GBM cell lines. (**A**,**B**) A1207 and U87 cells were treated with vehicle (DMSO) or CX-4945 (10 µM) for 72 h and the mitochondrial activity was analyzed by WST-1 assays. Vehicle-treated cells were set 100%. Mean ± SD. * *p* < 0.05 vs. vehicle (*n* = 3). (**C**,**D**) A1207 and U87 cells were treated as described in (**A**,**B**) and the MFI of BrdU-positive cells (% of total gated cells) was assessed by flow cytometry. Vehicle-treated cells were used as control. Mean ± SD. Mean ± SD. * *p* < 0.05 vs. vehicle (*n* = 7). (**E**,**F**) A1207 and U87 cells were seeded in a 24-well plate, cultivated for 24 h and subsequently treated with vehicle (DMSO) or CX-4945 (10 µM). The cell number was determined at 0 h, 24 h, 48 h and 72 h after treatment. The cell number at 0 h was set 100%. Mean ± SD. * *p* < 0.05 vs. vehicle (*n* = 4). (**G**–**J**) A1207 and U87 cells were cultivated and treated as described in (**E**,**F**) and analyzed by scratch assays (scale bars in G and I: 200 µm). The gap area was measured after 0 h and 24 h. The gap area at 0 h was set 100%. Mean ± SD. * *p* < 0.05 vs. vehicle at 24 h; ^#^
*p* < 0.05 vs. 0 h (*n* = 4). (**K**,**L**) A1207 and U87 cells were treated as described in (**A**,**B**) and the cytotoxicity of CX-4945 was assessed by LDH assays. Vehicle-treated cells were used as control and set 100% (*n* = 3).

**Figure 4 cancers-13-01678-f004:**
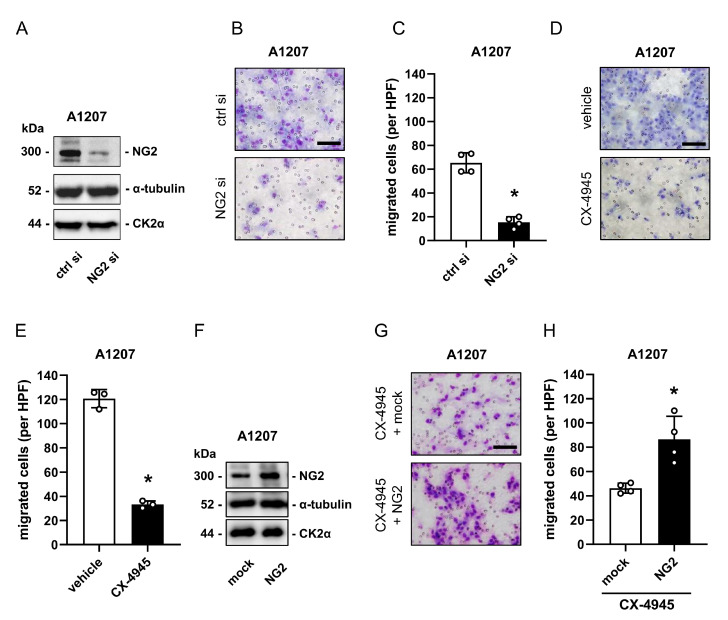
CK2 inhibition reduces the migratory capacity of NG2-positive GBM cell lines. (**A**) A1207 cells were transfected with ctrl si or NG2 si for 72 h. The cells were harvested, lysed and the expression of NG2, CK2α and α-tubulin (as loading control) was analyzed by western blot. (**B**) A1207 cells were treated as described in (**A**), detached and their migration was assessed by transwell assays (scale bar: 50 µm). (**C**) A1207 cells were treated as described in (**A**) and the migration was quantitatively assessed. Ctrl si-transfected cells were used as control. Mean ± SD. * *p* < 0.05 vs. ctrl si (*n* = 4). (**D**) A1207 cells were treated with vehicle (DMSO) or CX-4945 (10 µM) for 72 h, detached and their migration was assessed by transwell assays (scale bar: 50 µm). (**E**) A1207 cells were treated as described in (**D**) and migration was quantitatively assessed. Vehicle-treated cells were used as control. Mean ± SD. * *p* < 0.05 vs. vehicle (*n* = 4). (**F**) A1207 were transfected with mock (pEF6 vector) or NG2 plasmid and incubated for 48 h. Expression of NG2, CK2α and α-tubulin (as loading control) was analyzed by western blot. (**G**) A1207 cells were transfected as described in (**F**), treated with CX-4945 for 24 h, detached and their migration was assessed by transwell assays (scale bar: 50 µm). (**H**) A1207 cells were transfected and treated as described in (**G**) and migration was quantitatively assessed. Mock-transfected cells were used as control. Mean ± SD. * *p* < 0.05 vs. mock (*n* = 4).

**Figure 5 cancers-13-01678-f005:**
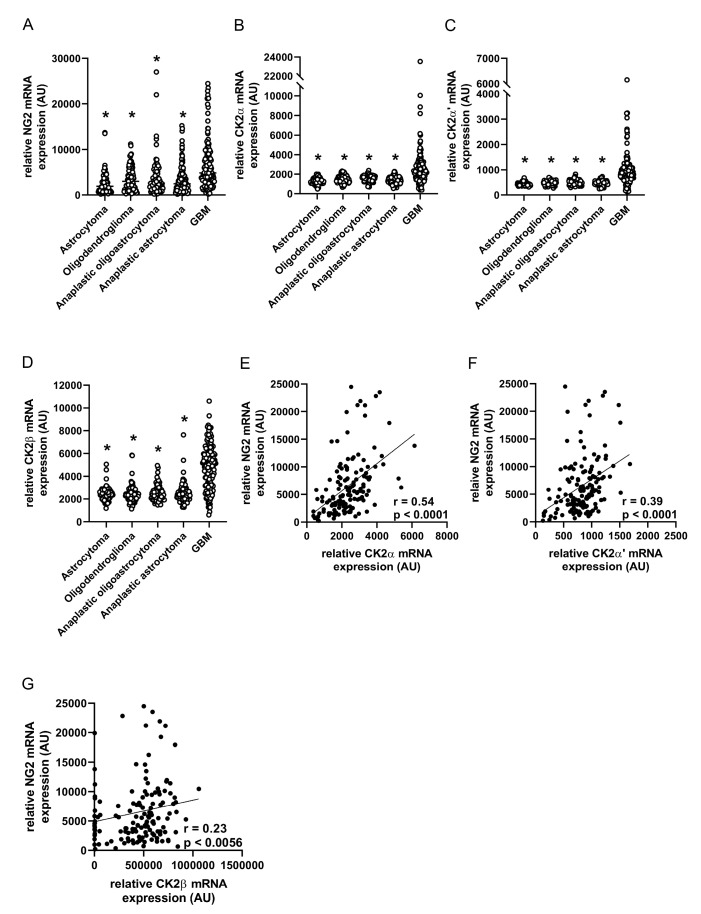
CK2 and NG2 mRNA expression positively correlate in human GBM. (**A**–**D**) The relative mRNA expression (TCGA-based data; arbitrary units (AU)) of NG2 (**A**), CK2α (**B**), CK2α’ (**C**) and CK2β (**D**) was analyzed in human astrocytoma (grade I-II), oligodendroglioma (grade II), anaplastic oligoastrocytoma and astrocytoma (grade III) as well as in GBM (grade IV). * *p* < 0.05 vs. GBM (*n* = 682). (**E**–**G**) Spearman correlations of NG2 mRNA expression (AU) with CK2α (**E**), CK2α’ (**F**) and CK2β (**G**) mRNA expression (TCGA-based data, AU, *n* = 141).

**Figure 6 cancers-13-01678-f006:**
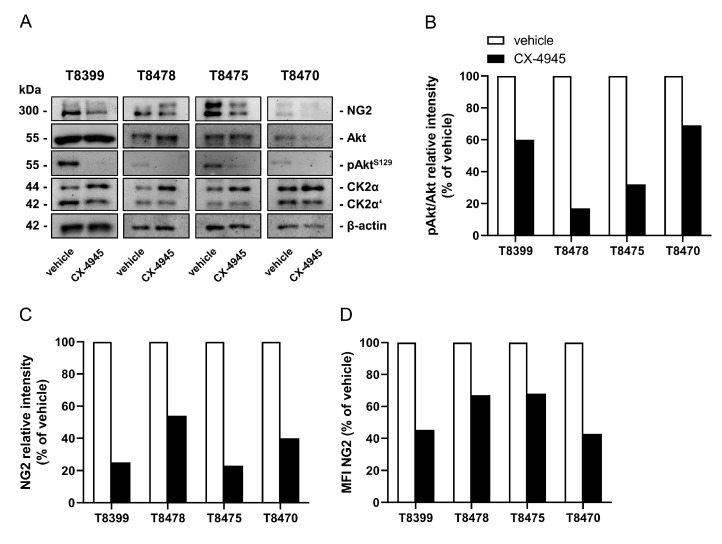
CK2 inhibition reduces NG2 expression in patient-derived GBM cells. (**A**) Patient-derived GBM cells (T8399, T8478, T8475 and T8470) were treated with vehicle (DMSO) or CX-4945 (10 µM) for 72 h. Subsequently, the cells were lysed and the expression of NG2, Akt, pAkt^S129^, CK2α, CK2α’ and β-actin (as loading control) was analyzed by western blot. (**B**,**C**) The cells were treated as described in (A) and the expression of pAkt/Akt (**B**) and NG2 (**C**) was quantitatively analyzed. Vehicle-treated cells were set 100%. (**D**) Patient-derived GBM cells (T8399, T8478, T8475 and T8470) were treated with vehicle (DMSO) or CX-4945 (10 µM) for 72 h, scratched and the MFI of NG2-positive cells was assessed by flow cytometry. Vehicle-treated cells were set 100%.

## Data Availability

Data are contained within the article or Appendix A.

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
