# Peer review of "CK2 Activity Mediates the Aggressive Molecular Signature of Glioblastoma Multiforme by Inducing Nerve/Glial Antigen (NG)2 Expression"

_cancers, 2021, doi:10.3390/cancers13071678_

Round 1
Reviewer 1 Report
Although the authors respond to my points in their report, the manuscript is not so different from the previous version. I think that a further effort should be done to improve its level, as detailed below:
- In response to my point 3, the authors performed experiments at lower times and CX-4945 concentrations. The results are shown in the report for the reviewer. They seem to indicate that 48h, instead of 72h, were probably enough, but this is not a problem, since there aren’t symptoms of toxicity at 72h; the point is that, in the revised manuscript, the authors include the methods used for these determinations, but not the results. I think they should be shown or at least discussed.
- In response to my point 4, on the lack of results obtained with different KO clones, the authors explain that they used a CRISPR-Cas9 system based on two tandem gRNAs, expected to minimize the off-target effects. They also mention that, by this approach, they generated several single cell-derived populations. However, they only show a CK2alpha western blot for two clones in the report for the reviewer, while the manuscript has remained unchanged. To convincingly exclude that the observed effects are related to a specific clone, the WBs of different KO clones should be shown. This applies to all results of Fig 1 I-N, where the individual WBs should be shown, besides the global quantification.
- In the response to my Minor point on the doublet bands appearing in some NG2 western blots, the authors mention the possibility of glycosylation or even of unknown phosphorylation events, but they don’t mention this point at all in the manuscript. Since the event is very clear and visible in different figures, I think it should be discussed, to inform the reader of possible explanations.
Author Response
Comment 1: In response to my point 3, the authors performed experiments at lower times and CX-4945 concentrations. The results are shown in the report for the reviewer. They seem to indicate that 48h, instead of 72h, were probably enough, but this is not a problem, since there aren’t symptoms of toxicity at 72h; the point is that, in the revised manuscript, the authors include the methods used for these determinations, but not the results. I think they should be shown or at least discussed.
Reply: According to the comment of the reviewer, we have included a novel paragraph in the revised version of our manuscript discussing the concentration of 10 µM CX-4945 used for our experiments. This paragraph reads as follows:
“We herein found that the inhibition of CK2 significantly reduces NG2 gene expression in the GBM cell lines A1207 and U87. For this purpose, CX-4945 was used to suppress CK2 activity. To identify suitable, non-toxic concentrations of this inhibitor, GBM cells were treated with 2.5 µM, 5 µM and 10 µM CX-4945 over 24 h, 48 h and 72 h, and both cell proliferation and cytotoxicity were determined. We found that none of the tested concentrations is cytotoxic. However, only 10 µM CX-4945 for 72 h lowered cell proliferation more than 50% (data not shown). More importantly, 10 µM CX-4945 was the most efficient concentration for the reduction of NG2 expression (data not shown). Based on these findings, we decided to use 10 µM CX-4945 to study the effects on NG2 expression and NG2-dependent cellular functions. To exclude that the herein observed reduced NG2 expression is due to unspecific effects of CX-4945 and, thus, CK2-independent, we additionally generated CK2α KO cell lines showing that the loss of CK2 activity significantly reduces NG2 expression.”
(see page 11, lines 9-21)
Comment 2: In response to my point 4, on the lack of results obtained with different KO clones, the authors explain that they used a CRISPR-Cas9 system based on two tandem gRNAs, expected to minimize the off-target effects. They also mention that, by this approach, they generated several single cell-derived populations. However, they only show a CK2alpha western blot for two clones in the report for the reviewer, while the manuscript has remained unchanged. To convincingly exclude that the observed effects are related to a specific clone, the WBs of different KO clones should be shown. This applies to all results of Fig 1 I-N, where the individual WBs should be shown, besides the global quantification.
Reply: According to the comment of the reviewer, we have included a supplemental figure showing Western blot analyses of additional CK2α KO clones (see page 3, lines 5-7 and supplemental figure S1A and B)
Comment 3: In the response to my Minor point on the doublet bands appearing in some NG2 western blots, the authors mention the possibility of glycosylation or even of unknown phosphorylation events, but they don’t mention this point at all in the manuscript. Since the event is very clear and visible in different figures, I think it should be discussed, to inform the reader of possible explanations
Reply: We agree with the reviewer that information about the higher molecular forms of NG2, as detected by Western blots, would be useful for the reader of this manuscript. Therefore, we included a novel paragraph in the discussion section of the revised manuscript version, which reads as follows:
“Various posttranslational modifications of the NG2 core protein, resulting in higher molecular forms, have been identified. For instance, the group of Stallcup showed that NG2 is phosphorylated at threonine 2256 by protein kinase C and threonine 2314 by ERK1/2, which promotes cell proliferation and cell motility [1]. Moreover, high molecular forms of NG2 caused by glycosylation are observed. However, the nature of this glycosylation and its biological significance is still unknown [2; 3]. In this study, we detected higher molecular forms of NG2 by Western blot, which are partially reduced after CK2 inhibition. Hence, we cannot exclude that CK2 additionally reduces NG2 protein level by phosphorylation of the proteoglycan or by inhibition of glycosyltransferase.”
(see page 11, lines 29-37)
References:
- Makagiansar IT, Williams S, Mustelin T, Stallcup WB (2007) Differential phosphorylation of NG2 proteoglycan by ERK and PKCalpha helps balance cell proliferation and migration. J Cell Biol 178: 155-165
- Girolamo F, Dallatomasina A, Rizzi M, Errede M, Walchli T, Mucignat MT, Frei K, Roncali L, Perris R, Virgintino D (2013) Diversified expression of NG2/CSPG4 isoforms in glioblastoma and human foetal brain identifies pericyte subsets. PLoS One 8: e84883
- Muir EM, Fyfe I, Gardiner S, Li L, Warren P, Fawcett JW, Keynes RJ, Rogers JH (2010) Modification of N-glycosylation sites allows secretion of bacterial chondroitinase ABC from mammalian cells. Journal of biotechnology 145: 103-110
Reviewer 2 Report
In my opinion the authors have fully addressed the comments of reviewer 3.Author Response
We thank the reviewer that all comments were answered to his satisfaction.
This manuscript is a resubmission of an earlier submission. The following is a list of the peer review reports and author responses from that submission.
Round 1
Reviewer 1 Report
The findings by Schmitt et al. for putative aggressive behavior in glioblastomas is of great interest in regard to potential targeted therapies. The paper is fairly easy to follow. One minor suggestion is to include the meaning of the abbreviation, MFI, in line 107 rather than waiting until line 412. The paper provoked questions which the authors can consider addressing. (1) Can the possibility that CK2's phosphorylation of SP1, a transcription factor, is mediating suppression of NG2 be discussed as it was in their earlier paper (ref 19 in this paper)? (2) Does inhibition of CK2 with subsequent suppression of NG2 affect the migration of normal astrocytes? (3) Do the levels of NG2 and/or CK2 change when cells are cultured in vitro? (4) Are there mutations (SNPs, indels, copy number changes, etc.) in genes encoding CK2 and NG2 and are there gains or losses in the chromosomal regions of these genes that are known to occur in glioblastomas?
Author Response
Reviewer comments:
Major points:
- Can the possibility that CK2's phosphorylation of SP1, a transcription factor, is mediating suppression of NG2 be discussed as it was in their earlier paper (ref 19 in this paper)?
Reply: According to the comment of the reviewer, we have included a novel paragraph in the discussion section of the revised manuscript about the putative role of SP1 in NG2 gene expression. This paragraph reads as follows:
‘The analysis of the gene regulatory mechanism revealed that this is caused by a decreased transcriptional activity of a 114 bp fragment close to the NG2 start codon. This region harbors a binding site for the transcription factor SP1, which can activate or repress gene expression [1]. It has been reported that posttranslational modifications of SP1, including phosphorylation, affect GBM cell proliferation and invasion [2; 3]. Of note, CK2 phosphorylates SP1 and inhibition of this kinase increases its DNA binding capacity [4; 5]. Hence, it is tempting to speculate that CK2 regulates NG2 gene expression via SP1-dependent phosphorylation in GBM cells.’
(see page 11, lines 8-16)
References:
- O'Connor L, Gilmour J, Bonifer C (2016) The Role of the Ubiquitously Expressed Transcription Factor Sp1 in Tissue-specific Transcriptional Regulation and in Disease. The Yale journal of biology and medicine 89: 513-525
- Kambe A, Yoshioka H, Kamitani H, Watanabe T, Baek SJ, Eling TE (2009) The cyclooxygenase inhibitor sulindac sulfide inhibits EP4 expression and suppresses the growth of glioblastoma cells. Cancer prevention research 2: 1088-1099
- Park MH, Ahn BH, Hong YK, Min do S (2009) Overexpression of phospholipase D enhances matrix metalloproteinase-2 expression and glioma cell invasion via protein kinase C and protein kinase A/NF-kappaB/Sp1-mediated signaling pathways. Carcinogenesis 30: 356-365
- Armstrong SA, Barry DA, Leggett RW, Mueller CR (1997) Casein kinase II-mediated phosphorylation of the C terminus of Sp1 decreases its DNA binding activity. J Biol Chem 272: 13489-13495
- Zhang S, Kim KH (1997) Protein kinase CK2 down-regulates glucose-activated expression of the acetyl-CoA carboxylase gene. Arch Biochem Biophys 338: 227-232
- Does inhibition of CK2 with subsequent suppression of NG2 affect the migration of normal astrocytes?
Reply: Mature astrocytes only migrate under pathophysiological conditions, including inflammation, tissue injury or tumorigenesis [1; 2]. Moreover, mature astrocytes do not express NG2 [3-6]. Therefore, CK2 inhibition can neither reduce NG2 gene expression in astrocytes nor affect NG2-mediated migration of astrocytes. However, CK2 inhibition may regulate the migration of astrocytes independently of NG2 expression. In fact, several studies have already reported that CK2 is involved in various signaling pathways regulating cell migration, such as the extracellular signal-regulated kinases (ERK), Wnt/β-catenin or phosphoinositide 3-kinase (PI3K)/Akt pathways [7; 8].
References:
- Zhan JS, Gao K, Chai RC, Jia XH, Luo DP, Ge G, Jiang YW, Fung YW, Li L, Yu AC (2017) Astrocytes in Migration. Neurochem Res 42: 272-282
- Yang C, Iyer RR, Yu AC, Yong RL, Park DM, Weil RJ, Ikejiri B, Brady RO, Lonser RR, Zhuang Z (2012) beta-Catenin signaling initiates the activation of astrocytes and its dysregulation contributes to the pathogenesis of astrocytomas. Proc Natl Acad Sci U S A 109: 6963-6968
- Zhu X, Bergles DE, Nishiyama A (2008) NG2 cells generate both oligodendrocytes and gray matter astrocytes. Development 135: 145-157
- Dimou L, Gallo V (2015) NG2-glia and their functions in the central nervous system. Glia 63: 1429-1451
- Huang W, Zhao N, Bai X, Karram K, Trotter J, Goebbels S, Scheller A, Kirchhoff F (2014) Novel NG2-CreERT2 knock-in mice demonstrate heterogeneous differentiation potential of NG2 glia during development. Glia 62: 896-913
- Nishiyama A, Yang Z, Butt A (2005) Astrocytes and NG2-glia: what's in a name? J Anat 207: 687-693
- Borgo C, Ruzzene M (2019) Role of protein kinase CK2 in antitumor drug resistance. Journal of experimental & clinical cancer research : CR 38: 287
- Chua MM, Ortega CE, Sheikh A, Lee M, Abdul-Rassoul H, Hartshorn KL, Dominguez I (2017) CK2 in Cancer: Cellular and Biochemical Mechanisms and Potential Therapeutic Target. Pharmaceuticals 10
- Do the levels of NG2 and/or CK2 change when cells are cultured in vitro?
Reply: We did not investigate whether our patient-derived primary GBM cells change their NG2 and CK2 expression patterns during cultivation. We are aware that these cells can undergo changes after extensive cultivation, as they potentially acquire multiple genetic and epigenetic alterations, lose the heterogeneity present in the parental tumor, and tend to lack tumor-initiating as well as multilineage differentiation capacity. In fact, Lee et al. [1] reported that cultivated primary GBM cells change their expression pattern and morphology, however, only after passage 3. Therefore, we used patient-derived GBM cells only until passage 2.
References:
- Lee J, Kotliarova S, Kotliarov Y, Li A, Su Q, Donin NM, Pastorino S, Purow BW, Christopher N, Zhang W et al (2006) Tumor stem cells derived from glioblastomas cultured in bFGF and EGF more closely mirror the phenotype and genotype of primary tumors than do serum-cultured cell lines. Cancer cell 9: 391-403
- Are there mutations (SNPs, indels, copy number changes, etc.) in genes encoding CK2 and NG2 and are there gains or losses in the chromosomal regions of these genes that are known to occur in glioblastomas?
Reply: According to the comment of the reviewer, we have included a novel paragraph in the discussion section of our revised manuscript addressing the relevance of NG2 and CK2 mutations in GBM. This paragraph reads as follows:
‘In GBM, overexpression of NG2 has not been reported to be a result of genetic aberrations, such as gene amplifications or chromosome translocation [1]. Moreover, there are no mutations known in the NG2 gene leading to gain or loss of function. In contrast, genome-wide copy number variation analyses in GBM demonstrated that chromosome 20 harbors frequent gains in gene dosage that may be driven by several oncogenic targets [2]. Of note, the CK2α gene is located on chromosome 20 and it has been reported that CK2 expression is required for activation of survival pathways, including the JAK/STAT, NFκB and PI3K/AKT pathways in GBM [3]. Hence, it can be assumed that the gains in CK2α gene dosage may be a crucial oncogenic driver during gliomagenesis.’
(see page 11, lines 52-54 and page 12, lines 1-7)
References:
- Ilieva KM, Cheung A, Mele S, Chiaruttini G, Crescioli S, Griffin M, Nakamura M, Spicer JF, Tsoka S, Lacy KE et al (2017) Chondroitin Sulfate Proteoglycan 4 and Its Potential As an Antibody Immunotherapy Target across Different Tumor Types. Front Immunol 8: 1911
- Bredel M, Scholtens DM, Harsh GR, Bredel C, Chandler JP, Renfrow JJ, Yadav AK, Vogel H, Scheck AC, Tibshirani R et al (2009) A network model of a cooperative genetic landscape in brain tumors. Jama 302: 261-275
- Zheng Y, McFarland BC, Drygin D, Yu H, Bellis SL, Kim H, Bredel M, Benveniste EN (2013) Targeting protein kinase CK2 suppresses prosurvival signaling pathways and growth of glioblastoma. Clin Cancer Res 19: 6484-6494
Minor point:
One minor suggestion is to include the meaning of the abbreviation, MFI, in line 107 rather than waiting until line 412.
Reply: According to the comment of the reviewer, we have introduced the abbreviation on page 4 in line 1.
Reviewer 2 Report
This is the interesting continuation of the authors’ previous work on the correlation between CK2 and NG2, here extended to glioblastoma (GBM) cells, also employing patient-derived cells.
The hypothesis of CK2 targeting in GBM has been already explored and proposed by other groups (partly cited by the Authors), with promising results. The novelty of this work is the focus on NG2-positive GBM, showing that CK2 inhibition reduces proliferation and migration in NG2-positive cells. However, the effect in NG2-negative cells in comparison has not been evaluated. While the results are quite convincing in showing that NG2 level is under the control of CK2 in GBM, additional experiments would reinforce the concept that “CK2 mediates the aggressive molecular signature of GBM by inducing NG2 expression” and that the efficacy of CK2 targeting in GBM is related to this CK2-NG2 axis.
Major points:
- To confirm the importance of NG2 in the pro-proliferation action of CK2 in GBM, CK2 inhibition by cell treatment with CX-4945 (as in Fig 3) should be performed also in NG2-negative cells, or in NG2-siRNA cells, expecting a lower response
- Similarly, in Fig 4 it would be important to know if the effect of CX-4945 on the migration capacity is reduced in NG2-siRNA cells
- The CX-4945 cell treatments for cell proliferation and migration have been performed at a single very high concentration (10uM) and for a single very long time (72h). Aren’t milder conditions effective? All data are based on the results with only one CK2 inhibitor. They have been confirmed by generating CK2alpha KO cells, but the gold standard, in case of KO cells, is to confirm results with at least two KO clones, generated with different guides. In the Methods, two single guide RNAs are indicated for the CK2alpha KO cells: where are the results with the second clone?
Minor points
- The NG2 band in some WBs appears as a doublet: see for example Fig 1A, U87 cells (but not same cells in Fig 1H), or patient-derived cells of Fig 6A. Moreover, in some patient-derived cells the upper band level is reduced, in others increased by the CX-4945 treatment. Do the author have any explanation? Is NG2 phosphorylated?
Author Response
Reviewer comments:
Major points:
- To confirm the importance of NG2 in the pro-proliferation action of CK2 in GBM, CK2 inhibition by cell treatment with CX-4945 (as in Fig 3) should be performed also in NG2-negative cells, or in NG2-siRNA cells, expecting a lower response.
Reply: We did not perform additional experiments with NG2-negative cells or NG2-siRNA-treated cells, because it is known that the loss of NG2 markedly affects cell proliferation [1]. In addition, it is also known that CK2 inhibition with CX-4945 reduces cell proliferation independently of NG2 [2]. Moreover, we found in the present study that inhibition of CK2 with CX-4945 reduces the gene expression of NG2. Hence, treatment with CX-4945 is comparable to NG2 gene silencing. Accordingly, it is to be expected that the treatment of NG2-siRNA-treated cells and NG2-positive cells with CX-4945 results in comparable proliferation rates.
References:
- Hsu SC, Nadesan P, Puviindran V, Stallcup WB, Kirsch DG, Alman BA (2018) Effects of chondroitin sulfate proteoglycan 4 (NG2/CSPG4) on soft-tissue sarcoma growth depend on tumor developmental stage. J Biol Chem 293: 2466-2475
- Siddiqui-Jain A, Drygin D, Streiner N, Chua P, Pierre F, O'Brien SE, Bliesath J, Omori M, Huser N, Ho C et al (2010) CX-4945, an orally bioavailable selective inhibitor of protein kinase CK2, inhibits prosurvival and angiogenic signaling and exhibits antitumor efficacy. Cancer Res 70: 10288-10298
- Similarly, in Fig 4 it would be important to know if the effect of CX-4945 on the migration capacity is reduced in NG2-siRNA cells.
We did not perform additional experiments with NG2-negative cells or NG2-siRNA-treated cells, because in the original version of our manuscript we already verified the function of NG2 in CK2-dependent migration. For this purpose, we overexpressed NG2 in GBM cells, which were subsequently treated with CX-4945. Thereafter, transmigration assays were performed (see manuscript Fig. 4F-H). This approach guaranteed that NG2 expression is driven by the promoter of the expression plasmid and not affected by CK2 activity [1]. We found that exogenous NG2 rescued the anti-migratory effect of CK2 inhibition in GBM cells. These experiments clearly demonstrate the importance of NG2 in the pro-migratory action of CK2 in GBM. We have additionally discussed these results in the revised version of our manuscript, which reads as follows:
‘In line with these findings, we could show that CK2 inhibition reduces the migration of NG2-positive GBM cell lines. It is well known that CK2 regulates cell migration via various signaling pathways [2; 3]. To verify that the herein observed anti-migratory effect is mediated by NG2, we performed additional rescue experiments. Our results clearly demonstrate the importance of NG2 in CK2-dependent migration of GBM cells, as shown by an improved migratory ability of NG2-overexpressing cells after CK2 inhibition.’
(see page 11, lines 21-27)
References:
- Schmitt BM, Boewe AS, Becker V, Nalbach L, Gu Y, Gotz C, Menger MD, Laschke MW, Ampofo E (2020) Protein Kinase CK2 Regulates Nerve/Glial Antigen (NG)2-Mediated Angiogenic Activity of Human Pericytes. Cells 9
- Rowse AL, Gibson SA, Meares GP, Rajbhandari R, Nozell SE, Dees KJ, Hjelmeland AB, McFarland BC, Benveniste EN (2017) Protein kinase CK2 is important for the function of glioblastoma brain tumor initiating cells. J Neurooncol 132: 219-229
- Zheng Y, McFarland BC, Drygin D, Yu H, Bellis SL, Kim H, Bredel M, Benveniste EN (2013) Targeting protein kinase CK2 suppresses prosurvival signaling pathways and growth of glioblastoma. Clin Cancer Res 19: 6484-6494
- The CX-4945 cell treatments for cell proliferation and migration have been performed at a single very high concentration (10uM) and for a single very long time (72h). Aren’t milder conditions effective?
Reply: Before we started our study, we determined the efficient concentration (EC)50 of CX-4945 by means of WST-1 assays. For this purpose, we treated A1207 and U87 cells with different concentrations of CX-4945 (2.5 µM, 5 µM and 10 µM) and analyzed the cell proliferation after 24 h, 48 h and 72 h. We found that only 10 µM of CX-4945 for 72 h lowers cell proliferation more than 50%, whereas milder conditions (in terms of concentration and time period) did not cross the EC50 threshold (Fig. 1A). Additional LDH assays revealed that none of the tested concentrations were cytotoxic (Fig. 1B). Moreover, 10 µM of CX-4945 was the most efficient concentration for the reduction of NG2 expression (< 50%) (Fig. 1C). Based on these findings, we treated the cells with 10 µM CX-4945 for 72 h to study effects on NG2 expression and NG2-dependent cellular functions.
Figure 1: (A and B) A1207 and U87 cells were treated with vehicle (DMSO) or different concentrations of CX-4945 (2.5 µM, 5 µM and 10 µM) for 24 h, 48 h and 72 h. The mitochondrial activity was analyzed by WST-1 assays (EC50 is marked by a blue broken line) (A) and the cytotoxicity was assessed by LDH assays (B). Vehicle-treated cells were used as control and set 100%. Mean ± SD, (n = 3). (C) Cells were treated as described in (A and B), scratched and the mean fluorescence intensity (MFI) of NG2-positive cells was assessed by flow cytometry. MFI of vehicle-treated cells was set 100%. Mean ± SD, (n = 3).
- All data are based on the results with only one CK2 inhibitor. They have been confirmed by generating CK2alpha KO cells, but the gold standard, in case of KO cells, is to confirm results with at least two KO clones, generated with different guides. In the Methods, two single guide RNAs are indicated for the CK2alpha KO cells: where are the results with the second clone?
Reply: In the present study, we used the CRISPR/Cas9-D10A all-in-one vector containing the Cas9D10A mutant nuclease as well as two tandem gRNAs [1]. The Cas9-D10A mutant nuclease produces single strand breaks with “sticky ends”, resulting in overhanging DNA termini. This is mediated by two tandem gRNAs in parallel, which guide the Cas9-D10A mutant nuclease to different target sequences in the CK2α gene. Thereby, off-target effects are minimized to undetectable levels while retaining high levels of on-target mutagenesis [1]. By using this system, we generated several single cell-derived populations and analyzed the CK2 protein amount by Western blot. CK2α was not detectable within these clones as shown
by Western blot analyses (Fig. 2).
Figure 2: A1207 wild type and CK2α KO cells (clone 1 and 2) were lysed and the expression of CK2α and α-tubulin (as loading control) was analyzed by Western blot.
References:
- Chiang TW, le Sage C, Larrieu D, Demir M, Jackson SP (2016) CRISPR-Cas9(D10A) nickase-based genotypic and phenotypic screening to enhance genome editing. Sci Rep 6: 24356
Minor points:
- The NG2 band in some WBs appears as a doublet: see for example Fig 1A, U87 cells (but not same cells in Fig 1H), or patient-derived cells of Fig 6A.
Reply: It is well known that higher molecular weights of NG2 detected by WB are due to glycosylated isoforms. In its fully glycosylated form, NG2 has an apparent molecular weight of >500 kDa and this isoform often coexists with less glycosylated variants running in the range of 300-350 kDa by SDS-PAGE [1; 2]. The nature of NG2 glycosylation and its biological significance is still unknown. One idea for NG2 glycosylation is that cancer cells produce a secretory form of NG2, which is only detectable during the phase of mitosis segregation to be thereafter released into the extracellular matrix [1].
References:
- Girolamo F, Dallatomasina A, Rizzi M, Errede M, Walchli T, Mucignat MT, Frei K, Roncali L, Perris R, Virgintino D (2013) Diversified expression of NG2/CSPG4 isoforms in glioblastoma and human foetal brain identifies pericyte subsets. PLoS One 8: e84883
- Muir EM, Fyfe I, Gardiner S, Li L, Warren P, Fawcett JW, Keynes RJ, Rogers JH (2010) Modification of N-glycosylation sites allows secretion of bacterial chondroitinase ABC from mammalian cells. Journal of biotechnology 145: 103-110
- Moreover, in some patient-derived cells the upper band level is reduced, in others increased by the CX-4945 treatment. Do the author have any explanation? Is NG2 phosphorylated?
Reply: We agree with the reviewer that glycosylated isoforms of NG2 differ between vehicle- and CX-4945-treated samples. The function of these glycosylated isoforms is still unknown. Hence, we can only speculate about the underlying mechanism. An idea is that CK2 regulates the glycosylation of NG2 by phosphorylation of glycosyltransferase. The inhibition of CK2 would therefore result in changed NG2 glycosylation patterns.
Besides glycosylation, NG2 is further posttranslationally modified by phosphorylation. The group of Stallcup showed that NG2 is phosphorylated at threonine 2256 by protein kinase (PK)C and threonine 2314 by the extracellular signal-regulated kinase (ERK)1/2, resulting in enhanced cell proliferation and cell motility [1]. The loss of these phosphorylation sites reduces the molecular weight of NG2 about ~2 kDa. These differences are not detectable in 7.5% SDS-PAGE. However, it cannot be excluded that NG2 harbors further unidentified phosphorylation sites.
References:
- Makagiansar IT, Williams S, Mustelin T, Stallcup WB (2007) Differential phosphorylation of NG2 proteoglycan by ERK and PKCalpha helps balance cell proliferation and migration. J Cell Biol 178: 155-165
Reviewer 3 Report
In the manuscript entitled "CK2 activity mediates the aggressive molecular signature of glioblastoma multiforme by inducing nerve/glial (NG)2 expression" Schmitt and colleagues have shown the role of CKS on (NG)2 expression in the glioblastoma cells. The authors designed an excellent study and used great techniques to investigate the role of CK2 on (NG)2 expression. My concern is mostly the novelty of the study as these findings are already published by this group in the manuscript PMCID: PMC6847167. So, they should justify the novelty of their study. I see that here the authors have used the CRISPER technique instead of iRNA that was used in their previous publication but using different techniques is not enough to increase the novelty unless otherwise, they are going to compare two techniques that it would be a method manuscript.
Author Response
Reviewer comments:
Major point:
My concern is mostly the novelty of the study as these findings are already published by this group in the manuscript PMCID: PMC6847167. So, they should justify the novelty of their study. I see that here the authors have used the CRISPER technique instead of iRNA that was used in their previous publication but using different techniques is not enough to increase the novelty unless otherwise, they are going to compare two techniques that it would be a method.
Reply: The PMID number (PMC6847167) is only a conference abstract (296 words) from 2019 and not an original article.